# A Generic Model of the Pseudo-Random Generator Based on Permutations Suitable for Security Solutions in Computationally-Constrained Environments

**DOI:** 10.3390/s19235322

**Published:** 2019-12-03

**Authors:** Tomislav Unkašević, Zoran Banjac, Milan Milosavljević

**Affiliations:** 1VLATACOM Institute, 11070 Belgrade, Serbia; zoran.banjac@vlatacom.com; 2Singidunum University, 11000 Belgrade, Serbia; mmilosavljevic@singidunum.ac.rs

**Keywords:** pseudo-random generator, security, wireless sensor networks, IoT, probability distribution, correlation, information leakage

## Abstract

Symmetric cryptography methods have an important role in security solutions design in data protection. In that context, symmetric cryptography algorithms and pseudo-random generators connected with them have strong influence on designed security solutions. In the computationally constrained environment, security efficiency is also important. In this paper we proposed the design of a new efficient pseudo-random generator parameterized by two pseudo-random sequences. By the probabilistic, information-theoretic and number theory methods we analyze characteristics of the generator. Analysis produced several results. We derived sufficient conditions, regarding parameterizing sequences, so that the output sequence has uniform distribution. Sufficient conditions under which there is no correlation between parameterizing sequences and output sequence are also derived. Moreover, it is shown that mutual information between the output sequence and parameterizing sequences tends to zero when the generated output sequence length tends to infinity. Regarding periodicity, it is shown that, with appropriately selected parameterizing sequences, the period of the generated sequence is significantly longer than the periods of the parameterizing sequences. All this characteristics are desirable regarding security applications. The efficiency of the proposed construction can be achieved by selection parameterizing sequences from the set of efficient pseudo-random number generators, for example, multiple linear feedback shift registers.

## 1. Introduction

The expansion of communication and network technologies, as well as technological advances in the design and implementation of microprocessor devices, have led to the ability to informational connecting different devices and creation of intelligent systems capable of monitoring and managing complex processes. Communication devices utilize the Internet infrastructure and protocols to create a world of connected devices, like Wireless Sensor Networks (WSN) and Internet of Things (IoT). This technological advancement enables the progress of many technological and life processes bringing to us smart cities, autonomous vehicles, robotization and intelligent robot behavior [1,2,3,4]. In that context, information security has a very important role in compromising the integrity and privacy of data in such an integrated world can cause serious damage, even to the level of a general disaster [5,6,7,8,9]. Therefore, in addition to security mechanisms incorporated into Internet protocols, additional security mechanisms incorporated into devices and systems are used to prevent unintended behavior. Moreover, a huge number of that type of devices (sensors, cameras, surveillance systems) need to work in real time fashion so that the defined security mechanisms do not disrupt system behavior. They must be designed in such a way that it is easy to implement them both in hardware and software and their application should not disrupt system behavior i.e., they must be efficient [10,11].

There are various IoT applications, connected with different types of sensors, that have become an integral part of our lives, and most of them can be classified in common areas such as smart healthcare services, smart home, intelligent transportation, smart grid, etc. However, as a consequence of mass deployment, many IoT challenges have arisen, such as limited processing capability and memory resources, large amount of data to transmit, different operating characteristics of hardware, and heterogeneous data and networks types [12,13,14,15,16]. Moreover, personal privacy, data confidentiality and integrity are also a great challenge of IoT that must be overcome, particularly for devices with limited resources and heterogeneous technologies [13,14,17,18,19]. Cryptography can be used to protect confidentiality (or secrecy) of data and communication. It can also be used to ensure the integrity (or accuracy) of information as well as for authentication (and non-repudiation) services [20]. An important point in the IoT world is that most IoT solutions have a “closed design”, so it is often very difficult or even impossible to incorporate additional security mechanisms after the production process is completed. On the other hand, as a consequence of the limited software and hardware resources of IoT devices, the suite of cryptographic algorithms that can be implemented is narrowing, so the right measure must be found between the desired level of security and implementation capabilities, which makes the security issue even more challenging [13,16]. Different cryptographic algorithms that offer roughly the same level of security may require different power and resource consumption, so you need to choose the right one, subject to the limitations of some specific IoT application and deployed hardware [19]. Given that public-key crypto algorithms, compared to symmetric crypto algorithms, have far greater power and resource consumption due to their high processing time [21], it is a natural choice to use a symmetric algorithm in IoT security solution design. Detailed analysis and comparison of symmetric block-type algorithms such as AES, RC6, Twofish, SPECK128, LEA, and ChaCha20-Poly1305 algorithms in IoT devices are given in [16]. On the other hand, stream or sequential symmetric key ciphers are typically faster than block-type. Block ciphers, in general, require more memory resources to encrypt/decrypt larger chunks (block) of data, while sequential ciphers usually take only one or a few bits at a time, they have relatively low memory requirements and therefore are suitable to implement in limited scenarios. Stream cryptography algorithms, as a subgroup of symmetric cryptography algorithms, are among the most common cryptography data protection techniques. The idea comes from Shannon’s one-time pad system, where instead of a random sequence of encryption bits, a series of bits obtained from a pseudo-random generator is used [20]. The sequence generated by the pseudo-random generator is used for plain-text encryption and its properties determine the security of the protected data. Therefore, the basic cryptography goal in stream cryptography systems is to design pseudo-random bit/symbol generators with good cryptographic characteristics. Many ideas have been implemented in the last fifty years, with more or less success.

One of the most popular and widely used pseudo-random sequence generator is the RC4 generator defined in 1987 by Ron Rivest. The description of the algorithm was revealed by reverse engineering of the RSA INC software [22], and the correctness of the algorithm description obtained was confirmed by Rivest himself [23,24].

The RC4 algorithm owes its popularity to its simplicity and ease of implementation in both software and hardware. The high popularity and applicability has attracted the attention of the cryptanalytic community. The results of a deep and thorough analysis of this algorithm led to the detection of a number of weaknesses of the algorithm. A comprehensive review of the weaknesses identified is given in [25] where the empirically detected weaknesses are theoretically proved as well as the original results of the authors of the article. The compromitation of this algorithm was additionally contributed by the implementation methods in security protocols, so that its use in security protocols has not been recommended since 2015 [26].

On the other hand, the beauty and elegance of the idea itself suggest the possibility of its exploitation.

Our work is aimed to define low complexity and efficient generic model of the pseudo-random generator that does not suffer from the weaknesses immanent to the RC4 and which is suitable for the implementation of the security solution in the computational constrained microprocessor environments, e.g., WSN and IoT. This paper defines a pseudo-random generator that can, in some way, be considered as a generalization of ideas related to RC4 because it uses the time varying permutations, sequences for permutation changing and addressing output element from the current generator state. In order to prove plausible cryptographic properties of the proposed pseudo-random generator different mathematical techniques are used to analyze probability distribution of the output sequence, correlation properties, information leakage between the state of the generator and output sequence and periodicity.

The paper is organized in four parts. After the first part containing motivation for this work and introduction in the second part we introduce necessary notation, describe proposed pseudo-random generator and his relationship to RC4 in brief. The third part contains the analysis of the generator, some comments and remarks. The fourth part contains a summary of the paper’s results.

## 2. Notation and Generating Algorithm Description

Let $I_{k}=\left\{0,1,\ldots ,k-1\right\}$. Then with $P_{k}$ we will denote a set of all bijections from $I_{k}$ to $I_{k}$, and as usual, its elements will be named permutations. Set $P_{k}$ is a totally ordered set by the, so called, lexicographic order and have exactly k! elements, where ! denotes factorial operation. Elements of the set $P_{k}$ we will denote by $\Pi_{1},\Pi_{2},\ldots ,\Pi_{k!}$.

By $P\left\{X\right\}$ we will denote the probability of set *X*.

Let $S=\left\{f_{0},f_{1},\ldots ,f_{m-1}\right\}\subseteq P_{k}$ be a set of permutations on $I_{k}$ with the following properties:For any $\Pi_{j}\in P_{k},\;j=1,2,\ldots ,k!,$ exists a number l>0 and set of integers $i_{1},i_{2},\ldots ,i_{l}\in I_{m}$ such that $\Pi_{j}=f_{i_{l}}\circ f_{i_{l-1}}\circ \ldots \circ \;f_{i_{1}}$, where ∘ is composition of functions.For all $p,q\in I_{m}$ if p≠q then $f_{p}\neq f_{q}$.$\Pi_{1}=I\in S$ where *I* is identical permutation.

Let ${\left\{A_{n}\right\}}_{n=1}^{\infty }$ and ${\left\{C_{n}\right\}}_{n=1}^{\infty }$ be a two sequences of independent identically distributed random variables with $P\left\{A_{n}=l\right\}=a_{l},\;l=0,1,...,k-1$ and $P\left\{C_{n}=l\right\}=c_{l},\;l=0,1,...,\left(m-1\right).$

Then we will define a pseudo-random sequence ${\left\{Z_{n}\right\}}_{n=1}^{\infty }$ with equations
(1)$$\begin{array}{cc} g_{0}=p & p\in P_{k}\\ g_{n+1}=f_{C_{n+1}}\circ g_{n} & n\geq 0\\ Z_{n+1}=g_{n+1}\left(A_{n+1}\right) & n\geq 0. \end{array}$$

From (Equation 1) generating algorithm for the sequence ${\left\{Z_{n}\right\}}_{n=1}^{\infty }$ is obvious. As a first step we will construct the sequence of permutations ${\left\{g_{n}\right\}}_{n=0}^{\infty }$, $g_{n}\in$
$P_{k}$ using sequence ${\left\{C_{n}\right\}}_{n=1}^{\infty }$ and element $Z_{n}$ of the sequence ${\left\{Z_{n}\right\}}_{n=1}^{\infty }$ is computed as a value of the function $g_{n}$ at the point $A_{n}$. Graphical presentation of the sequence generating process is given on the Figure 1.

Defined generator algorithm apply time-varying permutations as well as the RC4 but in RC4 fixed set of permutations, set of transpositions, is used. Graphical presentation of the RC4 algorithm is given on the Figure 2. where the summatios and numbers with denote reduction modulo 256.

In the defined generator case any set *S* which is generator of $P_{k}$ can be used. Sequences that are used in RC4 applied transposition determination and address of permutation table position corresponds to sequences ${\left\{C_{n}\right\}}_{n=1}^{\infty }$ and ${\left\{A_{n}\right\}}_{n=1}^{\infty }$ in our algorithm respectively. While the mentioned sequences from RC4 are precisely defined, in our case sequences ${\left\{C_{n}\right\}}_{n=1}^{\infty }$ and ${\left\{A_{n}\right\}}_{n=1}^{\infty }$ are arbitrarily chosen and the generator is parameterized by those two sequences. Later in the paper we define sufficient conditions for sequences ${\left\{C_{n}\right\}}_{n=1}^{\infty }$ and ${\left\{A_{n}\right\}}_{n=1}^{\infty }$ to achieve good pseudo-random and security properties.

## 3. Analysis of the Generator

For every pseudo-random generator it is necessary to analyze its properties regarding the possibilities of output sequence prediction or reconstruction of the generator initial state. In that sense desirable properties are uniform distribution of the output sequence, nonexistence of the correlation between the output sequence and elements of the generator, nonexistence of the output sequence auto-correlation and long period of the output sequence. These features are especially important for generators used in security solutions and lack any of them usually have serious consequences on the security of the system. Different examples can be found in [20,27].

### 3.1. Distribution of the Generated Sequence

Intuitively one can expect that ${\left\{Z_{n}\right\}}_{n=1}^{\infty }$ has a uniform distribution but relatively weak constraints demanded for the sequences ${\left\{A_{n}\right\}}_{n=1}^{\infty }$ and ${\left\{C_{n}\right\}}_{n=1}^{\infty }$ require formal proof for the expectations. By the next theorem we will show that ${\left\{Z_{n}\right\}}_{n=1}^{\infty }$ has an asymptotically uniform distribution.

**Theorem** **1.***If*$\sum_{i=1}^{k}a_{i}=1$*and*$a_{i}>0$*for all*$i\in I_{k},$$\sum_{i=0}^{m-1}c_{i}=1$*and*$c_{i}>0$*for all*$i\in I_{m},$*then pseudo-random sequence*${\left\{Z_{n}\right\}}_{n=1}^{\infty }$*has an asymptotically uniform distribution i.e.,*$$\left(\forall l\in I_{k}\right)\;\left(\lim_{n\to \infty }P\left\{Z_{n}=l\right\}=\frac{1}{k}\right)$$*for*$l\in I_{k}$.


The proof of the Theorem 1 will be derived in two steps. First, using Markov chains theory, we will show that sequence ${\left\{g_{n}\right\}}_{n=0}^{\infty }$ has asymptotically uniform distribution and after that, in the second step, using that result we will show the statement of the theorem.

**Proof.** First we will analyze the sequence of functions ${\left\{g_{n}\right\}}_{n=0}^{\infty }$. It is obvious that $g_{n}\in$$P_{k}$, as a consequence of $\left(P_{k},\circ \right)$ being group. Next we will show that
(2)$$\left(\forall i\in \left\{1,2,\ldots ,k!\right\}\right)\left(\lim_{n\to \infty }P\left\{g_{n}=\Pi_{i}\right\}=\frac{1}{k!}\right)$$To prove (Equation 2) we will observe the sequence ${\left\{g_{n}\right\}}_{n=0}^{\infty }$ as a stationary Markov chain over the set of states $P_{k}$. Indeed, according to the definition of ${\left\{g_{n}\right\}}_{n=0}^{\infty }$, transition from the $g_{n}$ to $g_{n+1}$ doesn’t depend on the history of $g_{n}$, but only on the current state $g_{n}$ and the value of $C_{n}$.Denote by $G_{n}={\left[P\left\{g_{n}=\Pi_{i}\right\}\right]}_{1\times k!}$ a row matrix whose elements are the probabilities that after *n* steps the chain is in the state $\Pi_{i}$. Let $t_{i\;j}$ be the probability that the chain changes state from $\Pi_{i}$ to $\Pi_{j}$ in one step and $T={\left[t_{i\;j}\right]}_{k!\times k!}$ be one step transition matrix for the Markov chain. Denote by $T_{n}={\left[t_{i\;j}^{n}\right]}_{k!\times k!}$*n*-step probability transition matrix of that system starting at the state $\Pi_{i}$ changes to state $\Pi_{j}$ after exactly *n* steps. It is well known, (see [28,29]), that $T_{n}=T^{n}$ and that,
(3)$$\begin{array}{ccc} G_{n} & = & G_{0}T_{n}\\ \lim_{n\to \infty }G_{n} & = & \lim_{n\to \infty }G_{0}T_{n}=G_{0}\lim_{n\to \infty }T_{n} \end{array}$$When the limit values in (Equation 3) exists. To show that $\lim_{n\to \infty }T_{n}$ exists it is sufficient to show that such $n_{0}\in N$ exists for which $t_{i\;j}^{n_{0}}>0$ for all $i,j\in \left\{1,2,...,k!\right\}$ (see [28,29]).Let us define numbers $n_{i\;j}$ as
(4)$$n_{i\;j}=\min_{r}\left\{r\;|\;\left(\exists \left(i_{1},i_{2},\ldots ,i_{r}\right)\in I_{m}^{r}\right)\left(f_{i_{r}}\circ \ldots \circ \;f_{i_{2}}\circ f_{i_{1}}=\Pi_{j}\circ \Pi_{i}^{-1}\right)\right\}.$$Due to the properties of the set *S* it is clear that $n_{i\;j}>0$. Let $n_{0}$ be $\max_{i,j\in \left\{1,2,\ldots ,k!\right\}}n_{i\;j}$ and show that $t_{i\;j}^{n_{0}}$ is greater than zero. Because $\left(P_{k},\circ \right)$ is group then the equation $x\circ \Pi_{i}=\Pi_{j}$ has exactly one solution, $\Pi_{j}\circ \Pi_{i}^{-1}$, so we can write
(5)$$\begin{array}{cc} t_{i\;j}^{n_{0}} & =\sum_{\left(i_{1},\ldots ,i_{n_{0}}\right)}P\left\{f_{i_{n_{0}}}\circ \ldots \circ f_{i_{2}}\circ f_{i_{1}}\circ \Pi_{i}=\Pi_{j}\right\}\\ \; & =\sum_{\left(i_{1},\ldots ,i_{n_{0}}\right)}P\left\{f_{i_{n_{0}}}\circ \ldots \circ f_{i_{2}}\circ f_{i_{1}}=\Pi_{j}\circ \Pi_{i}^{-1}\right\}\\ \; & =\sum_{\begin{array}{c} \left(i_{1},\ldots ,i_{n_{0}}\right)\\ f_{i_{n_{0}}}\circ \ldots \circ f_{i_{2}}\circ f_{i_{1}}=\Pi_{i}^{-1}\circ \Pi_{j} \end{array}}\prod_{l=1}^{n_{0}}P\left\{C_{l}=i_{l}\right\} \end{array}$$Now, to prove $t_{i\;j}^{n_{0}}>0$ it is sufficient to show that at least one summand in (Equation 5) is greater then zero, see [28,29]. Because $n_{i\;j}>0$ we can find a set of indices $\left\{i_{1},i_{2},\ldots ,i_{n_{i\;j}}\right\},\;n_{ij}\leq n_{0}$ such that $f_{i_{n_{ij}}}\circ \ldots \circ f_{i_{2}}\circ f_{i_{1}}=\Pi_{i}^{-1}\circ \Pi_{j}$. Because the index of identical permutation is 1, then the summand which corresponds to the set of indices $\left\{i_{1},i_{2},\ldots ,i_{n_{i\;j}},\underset{n_{0}-n_{i\;j}}{\underset{︸}{1,1,\ldots ,1}}\right\}$ is evidently greater then zero and we showed that $\lim_{n\to \infty }T_{n}$ exists. Because convergence is component wise, $\lim_{n\to \infty }t_{i\;j}^{n}=t_{j}^{*}$ exists. Limit value $\lim_{n\to \infty }T_{n}$ can be determined as a solution of the system of a equations known as Champman-Kolmogorov equations.
(6)$$\begin{array}{c} \sum_{j=1}^{k!}t_{j}^{*}t_{j\;l}=t_{l}^{*}\\ \sum_{j=1}^{k!}t_{j}^{*}=1 \end{array}$$
It is easy to check, by substitution, that the $t_{1}^{*}=$$t_{2}^{*}=\ldots =$$t_{k!}^{*}=\frac{1}{k!}$ is unique solution of the system (Equation 6).From now on the proof is straightforward. Let $l\in I_{k}$ be arbitrary, then
(7)$$\begin{array}{cc} P\left\{Z_{n}=l\right\} & =P\left\{g_{n}\left(A_{n}\right)=l\right\}\\ \; & =\sum_{i=1}^{k!}P\left\{g_{n}\left(A_{n}\right)=l\;|\;g_{n}=\Pi_{i}\right\}\;P\left\{g_{n}=\Pi_{i}\right\}\\ \; & =\sum_{i=1}^{k!}P\left\{\Pi_{i}\left(A_{n}\right)=l|\;g_{n}=\Pi_{i}\right\}\;P\left\{g_{n}=\Pi_{i}\right\}\\ \; & =\sum_{i=1}^{k!}\sum_{j=0}^{k-1}P\left\{\Pi_{i}\left(j\right)=l\;|\;A_{n}=j\right\}\;P\left\{A_{n}=j\right\}P\left\{g_{n}=\Pi_{i}\right\}. \end{array}$$Because $\left\{\Pi_{i}\left(j\right)=l\;|\right\}$ and $\left\{\;A_{n}=j\right\}$ are independent random variables it follows from (Equation 7) that
(8)$$\begin{array}{cc} P\left\{Z_{n}=l\right\} & =\sum_{i=1}^{k!}\sum_{j=0}^{k-1}P\left\{\Pi_{i}\left(j\right)=l\;|\;A_{n}=j\right\}\;P\left\{A_{n}=j\right\}P\left\{g_{n}=\Pi_{i}\right\}\\ \; & =\sum_{i=1}^{k!}\sum_{j=0}^{k-1}P\left\{\Pi_{i}\left(j\right)=l\right\}\;P\left\{A_{n}=j\right\}P\left\{g_{n}=\Pi_{i}\right\}\\ \; & =\sum_{j=0}^{k-1}P\left\{A_{n}=j\right\}\sum_{i=1}^{k!}\;P\left\{g_{n}=\Pi_{i}\right\}P\left\{\Pi_{i}\left(j\right)=l\right\} \end{array}$$Finding limit of the both sides of the (Equation 8) it follows
(9)$$\begin{array}{cc} \lim_{n\to \infty }P\left\{Z_{n}=l\right\} & =\lim_{n\to \infty }\sum_{j=0}^{k-1}P\left\{A_{n}=j\right\}\sum_{i=1}^{k!}\;P\left\{g_{n}=\Pi_{i}\right\}P\left\{\Pi_{i}\left(j\right)=l\right\}\\ \; & =\sum_{j=0}^{k-1}P\left\{A_{n}=j\right\}\sum_{i=1}^{k!}\;P\left\{\Pi_{i}\left(j\right)=l\right\}\lim_{n\to \infty }P\left\{g_{n}=\Pi_{i}\right\} \end{array}$$
because $P\left\{A_{n}=j\right\},\;P\left\{\Pi_{i}\left(j\right)=l\right\}$ do not depend on n. Using that $\lim_{n\to \infty }$P$\left\{g_{n}=\Pi_{i}\right\}$=$\frac{1}{k!}$ from (Equation 9) it follows that
$$\begin{array}{cc} \lim_{n\to \infty }P\left\{Z_{n}=l\right\} & =\sum_{j=0}^{k-1}P\left\{A_{n}=j\right\}\sum_{i=1}^{k!}\;P\left\{\Pi_{i}\left(j\right)=l\right\}\lim_{n\to \infty }P\left\{g_{n}=\Pi_{i}\right\}\\ \; & =\frac{1}{k!}\sum_{j=0}^{k-1}P\left\{A_{n}=j\right\}\sum_{i=1}^{k!}\;P\left\{\Pi_{i}\left(j\right)=l\right\}\\ \; & =\frac{1}{k!}\sum_{j=0}^{k-1}P\left\{A_{n}=j\right\}\left(k-1\right)!\\ \; & =\frac{\left(k-1\right)!}{k!}\sum_{j=0}^{k-1}P\left\{A_{n}=j\right\}\\ \; & =\frac{1}{k} \end{array}$$
which proves the theorem. □

**Remark** **1.**
*Asymptotically uniform distribution of the sequence $Z_{n},\;n=1,2,...\ldots ,$ as we have shown, is a consequence of the asymptotically uniform distribution of ${\left\{g_{n}\right\}}_{n=0}^{\infty }.$ If $G_{0}={\left[\frac{1}{k!},\frac{1}{k!},\ldots ,{\frac{1}{k!}}_{i}\right]}_{1\times k!}$ it is easy to verify that ${\left\{g_{n}\right\}}_{n=0}^{\infty }$ has a uniform distribution and as a consequence $Z_{n},\;n=1,2,\ldots$ has a uniform distribution too.*


**Theorem** **2.**
*If the random variables $A_{n},\;n=1,2,...\ldots$ are uniformly distributed then for all $z\in I_{k}$*
$$P\left\{Z_{n}=z\right\}=\frac{1}{k}$$


**Proof.** By the generator definition we have
(10)$$\begin{array}{cc} P\left\{Z_{n}=z\right\} & =\sum_{i=1}^{k!}\sum_{j=0}^{k-1}P\left\{Z_{n}=z\;|\;g_{n}=\Pi_{i}\wedge A_{n}=j\right\}\cdot P\left\{\;g_{n}=\Pi_{i}\wedge A_{n}=j\right\}\\ \; & =\sum_{i=1}^{k!}\sum_{j=0}^{k-1}P\left\{\Pi_{i}\left(j\right)=z\right\}\cdot P\left\{\;g_{n}=\Pi_{i}\right\}\cdot P\left\{A_{n}=j\right\} \end{array}$$
because $A_{n}$ is uniformly distributed from (Equation 10) it follows that
$$\begin{array}{cc} P\left\{Z_{n}=z\right\} & =\sum_{i=1}^{k!}\sum_{j=0}^{k-1}P\left\{\Pi_{i}\left(j\right)=z\right\}\cdot P\left\{\;g_{n}=\Pi_{i}\right\}\cdot \frac{1}{k}\\ \; & =\frac{1}{k}\sum_{i=1}^{k!}\sum_{j=0}^{k-1}P\left\{\Pi_{i}\left(j\right)=z\right\}\cdot P\left\{\;g_{n}=\Pi_{i}\right\}\\ \; & =\frac{1}{k}\sum_{i=1}^{k!}P\left\{\;g_{n}=\Pi_{i}\right\}\sum_{j=0}^{k-1}P\left\{\Pi_{i}\left(j\right)=z\right\}\\ \; & =\frac{1}{k} \end{array}$$
because $\sum_{i=1}^{k!}P\left\{\;g_{n}=\Pi_{i}\right\}=1$ and $\sum_{j=0}^{k-1}P\left\{\Pi_{i}\left(j\right)=z\right\}=1$ and theorem is proved.□

By these theorems we showed that generator has at least asymptotically uniform distribution of the values in the generator output sequence. This is important security feature because it indicates impossibility of the prediction of the generator output sequence based on the probability distribution of the output sequence values.

#### Correlation Properties

**Theorem** **3.**
*If the random variables $A_{n},\;n=1,2,...\ldots$ are uniformly distributed then for all $a,b\in I_{k},$*

$P\left\{Z_{n+l}=b\wedge Z_{n}=a\right\}=\frac{1}{k^{2}}$

$P\left\{Z_{n+l}=b|Z_{n}=a\right\}=\frac{1}{k}$



**Proof.** 
By the generator definition we have
(11)$$\begin{array}{cc} \; & P\left\{Z_{n+l}=b\wedge Z_{n}=a\right\}=P\left\{g_{n+k}\left(A_{n+k}\right)=b\wedge g_{n}\left(A_{n}\right)=a\right\}=\\ \; & =\sum_{i=1}^{k!}\sum_{j=1}^{k!}P\left\{g_{n+k}\left(A_{n+k}\right)=b\wedge g_{n}\left(A_{n}\right)=a\;|\;g_{n+k}=\Pi_{i}\wedge g_{n}=\Pi_{j}\right\}\\ \; & \;\;\;\;\cdot P\left\{g_{n+k}=\Pi_{i}\wedge g_{n}=\Pi_{j}\right\}\\ \; & =\sum_{i=1}^{k!}\sum_{j=1}^{k!}P\left\{\Pi_{i}\left(A_{n+k}\right)=b\wedge \Pi_{j}\left(A_{n}\right)=a\;\right\}\cdot P\left\{g_{n+k}=\Pi_{i}\wedge g_{n}=\Pi_{j}\right\}\\ \; & =\sum_{i=1}^{k!}\sum_{j=1}^{k!}P\left\{\Pi_{i}\left(A_{n+k}\right)=b\wedge \Pi_{j}\left(A_{n}\right)=a\;\right\}\cdot P\left\{g_{n+k}=\Pi_{i}\;|\;g_{n}=\Pi_{j}\right\}\\ \; & \;\;\;\;\cdot P\left\{g_{n}=\Pi_{j}\right\}. \end{array}$$
Now, using notation from the Theorem 1 $P\left\{g_{n+k}=\Pi_{i}\;|\;g_{n}=\Pi_{j}\right\}$ is equal to $t_{i,j}^{k}$ and putting it in (Equation 11) it follows that
(12)$$\begin{array}{cc} \; & P\left\{Z_{n+l}=b\wedge Z_{n}=a\right\}=\\ \; & =\sum_{i=1}^{k!}\sum_{j=1}^{k!}P\left\{\Pi_{i}\left(A_{n+k}\right)=b\wedge \Pi_{j}\left(A_{n}\right)=a\;\right\}\cdot P\left\{g_{n+k}=\Pi_{i}\;|\;g_{n}=\Pi_{j}\right\}\\ \; & \;\;\;\;\cdot P\left\{g_{n}=\Pi_{j}\right\}\\ \; & =\sum_{i=1}^{k!}\sum_{j=1}^{k!}P\left\{\Pi_{i}\left(A_{n+k}\right)=b\wedge \Pi_{j}\left(A_{n}\right)=a\;\right\}\cdot t_{i,j}^{k}\cdot P\left\{g_{n}=\Pi_{j}\right\}. \end{array}$$
Because $\Pi_{i},\Pi_{j}$ are permutations $P\left\{\Pi_{i}\left(A_{n+k}\right)=b\wedge \Pi_{j}\left(A_{n}\right)=a\;\right\}$ is equal to $P\left\{A_{n+k}=\Pi_{i}^{-1}\left(b\right)\wedge A_{n}=\Pi_{j}^{-1}\left(a\;\right)\right\}$ and using that $A_{n+k},A_{n}$ are independent random variables from (Equation 12) it follows that
(13)$$\begin{array}{cc} \; & P\left\{Z_{n+l}=b\wedge Z_{n}=a\right\}=\\ \; & =\sum_{i=1}^{k!}\sum_{j=1}^{k!}P\left\{\Pi_{i}\left(A_{n+k}\right)=b\wedge \Pi_{j}\left(A_{n}\right)=a\;\right\}\cdot t_{i,j}^{k}\cdot P\left\{g_{n}=\Pi_{j}\right\}\\ \; & =\sum_{i=1}^{k!}\sum_{j=1}^{k!}P\left\{A_{n+k}=\Pi_{i}^{-1}\left(b\right)\wedge A_{n}=\Pi_{j}^{-1}\left(a\;\right)\right\}\cdot t_{i,j}^{k}\cdot P\left\{g_{n}=\Pi_{j}\right\}\\ \; & =\sum_{i=1}^{k!}\sum_{j=1}^{k!}P\left\{A_{n+k}=\Pi_{i}^{-1}\left(b\right)\right\}\cdot P\left\{A_{n}=\Pi_{j}^{-1}\left(a\;\right)\right\}\cdot t_{i,j}^{k}\cdot P\left\{g_{n}=\Pi_{j}\right\} \end{array}$$
Now, using that $A_{n+k}$ and $A_{n}$ are uniformly distributed independent random variables it follows from (Equation 13) that
(14)$$\begin{array}{cc} P & \left\{Z_{n+l}=b\wedge Z_{n}=a\right\}=\\ \; & =\sum_{i=1}^{k!}\sum_{j=1}^{k!}P\left\{A_{n+k}=\Pi_{i}^{-1}\left(b\right)\right\}\cdot P\left\{A_{n}=\Pi_{j}^{-1}\left(a\;\right)\right\}\cdot t_{i,j}^{k}\cdot P\left\{g_{n}=\Pi_{j}\right\}\\ \; & =\sum_{i=1}^{k!}\sum_{j=1}^{k!}\frac{1}{k}\cdot \frac{1}{k}\cdot t_{i,j}^{k}\cdot P\left\{g_{n}=\Pi_{j}\right\}\\ \; & =\frac{1}{k^{2}}\sum_{i=1}^{k!}P\left\{g_{n}=\Pi_{i}\right\}\sum_{j=1}^{k!}t_{i,j}^{k}\\ \; & =\frac{1}{k^{2}} \end{array}$$
because $\sum_{j=1}^{k!}t_{i,j}^{k}=1$ and $\sum_{i=1}^{k!}P\left\{g_{n}=\Pi_{i}\right\}=1,$ and statement is proved.Using statement of the Theorem 2 that $\;P\left\{Z_{n}=a\right\}=\frac{1}{k}$ by the definition of conditional probability it follows that
$$P\left\{Z_{n+l}=b|Z_{n}=a\right\}=\frac{P\left\{Z_{n+k}=b\wedge Z_{n}=a\right\}}{P\left\{Z_{n}=a\right\}}=\frac{\frac{1}{k^{2}}}{\frac{1}{k}}=\frac{1}{k}$$
which proves the statement. □


### 3.2. Information Leakage

Information leakage means existence of the correlation between generator output sequence and elements of its inner state. Such type correlation may be base for the process of reconstruction of the some generator state during his generation history. Knowledge of the one state during the generator work and knowledge of the generator algorithm allows prediction of the future elements of the output sequence which is undesirable in security applications. In this part, correlation with the state element sequences ${\left\{A_{n}\right\}}_{n=1}^{\infty }$ and ${\left\{C_{n}\right\}}_{n=1}^{\infty }$ with ${\left\{Z_{n}\right\}}_{n=1}^{\infty }$ is considered.

**Theorem** **4.**
*Under the conditions of the Theorem 1 we have*

*$\lim_{n\to \infty }\;P\left(Z_{n}=z|C_{n}=c\right)=\frac{1}{k}$*

$\lim_{n\to \infty }\;I\left(Z_{n},C_{n}\right)=0$

*where*
$z\in I_{k}$
*and*
$c\in I_{m}$


**Proof.** 
By the definition
(15)$$\begin{array}{cc} P & \left(Z_{n}=z|C_{n}=c\right)=\;\\ \; & =\frac{P\left(Z_{n}=z\wedge C_{n}=c\right)}{P\left(C_{n}=c\right)}=\frac{P\left(g_{n}\left(A_{n}\right)=z\wedge C_{n}=c\right)}{P\left(C_{n}=c\right)}\\ \; & =\frac{P\left(\left(\overset{k!}{\underset{i=1}{⋃}}g_{n}=\Pi_{i}\wedge \Pi_{i}\left(A_{n}\right)=z\right)\wedge C_{n}=c\right)}{P\left(C_{n}=c\right)}\\ \; & =\frac{P\left(\left(\overset{k!}{\underset{i=1}{⋃}}f_{C_{n}}\circ g_{n-1}=\Pi_{i}\wedge \Pi_{i}\left(A_{n}\right)=z\right)\wedge C_{n}=c\right)}{P\left(C_{n}=c\right)}\\ \; & =\frac{P\left(\overset{k!}{\underset{i=1}{⋃}}\left(\left(f_{C_{n}}\circ g_{n-1}=\Pi_{i}\wedge \Pi_{i}\left(A_{n}\right)=z\right)\wedge C_{n}=c\right)\right)}{P\left(C_{n}=c\right)}. \end{array}$$
Because $\left\{f_{C_{n}}\circ g_{n-1}=\Pi_{i}\right\}$ and $\left\{f_{C_{n}}\circ g_{n-1}=\Pi_{j}\right\}$ are disjoint events for $i,j\in I_{k}$ and i≠j, from (Equation 15) it follows that
(16)$$\begin{array}{cc} P & \left(Z_{n}=z|C_{n}=c\right)=\frac{P\left(\overset{k!}{\underset{i=1}{⋃}}\left(\left(f_{C_{n}}\circ g_{n-1}=\Pi_{i}\wedge \Pi_{i}\left(A_{n}\right)=z\right)\wedge C_{n}=c\right)\right)}{P\left(C_{n}=c\right)}\\ \; & =\frac{\sum_{i=1}^{k!}P\left(\left(\left(f_{C_{n}}\circ g_{n-1}=\Pi_{i}\wedge \Pi_{i}\left(A_{n}\right)=z\right)\wedge C_{n}=c\right)\right)}{P\left(C_{n}=c\right)} \end{array}$$
Now, using Bayes theorem from (Equation 16) it follows that
(17)$$\begin{array}{cc} P & \left(Z_{n}=z|C_{n}=c\right)=\frac{\sum_{i=1}^{k!}P\left(\left(\left(f_{C_{n}}\circ g_{n-1}=\Pi_{i}\wedge \Pi_{i}\left(A_{n}\right)=z\right)\wedge C_{n}=c\right)\right)}{P\left(C_{n}=c\right)}\\ \; & =\frac{\sum_{i=1}^{k!}\sum_{j=0}^{k-1}P\left(\left(\left(f_{C_{n}}\circ g_{n-1}=\Pi_{i}\wedge \Pi_{i}\left(A_{n}\right)=z\right)\wedge C_{n}=c\right)|\left(A_{n}=j\right)\right)\cdot P\left(A_{n}=j\right)}{P\left(C_{n}=c\right)}\\ \; & =\frac{\sum_{i=1}^{k!}\sum_{j=0}^{k-1}P\left(\left(\left(f_{C_{n}}\circ g_{n-1}=\Pi_{i}\wedge \Pi_{i}\left(j\right)=z\right)\wedge C_{n}=c\right)\right)\cdot P\left(A_{n}=j\right)}{P\left(C_{n}=c\right)}\\ \; & =\frac{\sum_{i=1}^{k!}\sum_{j=0}^{k-1}P\left(\left(g_{n-1}=f_{C_{n}}^{-1}\circ \Pi_{i}\wedge C_{n}=c\right)\wedge \Pi_{i}\left(j\right)=z\right)\cdot P\left(A_{n}=j\right)}{P\left(C_{n}=c\right)}. \end{array}$$
Now, because events $\;\left\{g_{n-1}=f_{C_{n}}^{-1}\circ \Pi_{i}\wedge C_{n}=c\right\}$ and $\left\{\Pi_{i}\left(j\right)=z\right\}$ are independent from (Equation 17) it follows that
(18)$$\begin{array}{cc} P & \left(Z_{n}=z|C_{n}=c\right)=\\ \; & =\frac{\sum_{i=1}^{k!}\sum_{j=0}^{k-1}P\left(\left(g_{n-1}=f_{C_{n}}^{-1}\circ \Pi_{i}\wedge C_{n}=c\right)\wedge \Pi_{i}\left(j\right)=z\right)\cdot P\left(A_{n}=j\right)}{P\left(C_{n}=c\right)}\\ \; & =\frac{\sum_{i=1}^{k!}\sum_{j=0}^{k-1}P\left(g_{n-1}=f_{C_{n}}^{-1}\circ \Pi_{i}\wedge C_{n}=c\right)\cdot P\left(\Pi_{i}\left(j\right)=z\right)\cdot P\left(A_{n}=j\right)}{P\left(C_{n}=c\right)} \end{array}$$
Grouping summands which depends on *j* from (Equation 18) it follows that
(19)$$\begin{array}{cc} P & \left(Z_{n}=z|C_{n}=c\right)=\\ \; & =\frac{\sum_{i=1}^{k!}\sum_{j=0}^{k-1}P\left(g_{n-1}=f_{C_{n}}^{-1}\circ \Pi_{i}\wedge C_{n}=c\right)\cdot P\left(\Pi_{i}\left(j\right)=z\right)\cdot P\left(A_{n}=j\right)}{P\left(C_{n}=c\right)}\\ \; & =\sum_{i=1}^{k!}\frac{P\left(g_{n-1}=f_{C_{n}}^{-1}\circ \Pi_{i}\wedge C_{n}=c\right)\sum_{j=0}^{k-1}P\left(A_{n}=j\right)\cdot P\left(\Pi_{i}\left(j\right)=z\right)}{P\left(C_{n}=c\right)}\\ \; & =\sum_{i=1}^{k!}\frac{P\left(g_{n-1}=f_{C_{n}}^{-1}\circ \Pi_{i}\wedge C_{n}=c\right)}{P\left(C_{n}=c\right)}\sum_{j=0}^{k-1}P\left(A_{n}=j\right)\cdot P\left(\Pi_{i}\left(j\right)=z\right)\\ \; & =\sum_{i=1}^{k!}P\left(g_{n-1}=f_{C_{n}}^{-1}\circ \Pi_{i}|C_{n}=c\right)\sum_{j=0}^{k-1}P\left(A_{n}=j\right)\cdot P\left(\Pi_{i}\left(j\right)=z\right)\\ \; & =\sum_{i=1}^{k!}P\left(g_{n-1}=f_{c}^{-1}\circ \Pi_{i}\right)\sum_{j=0}^{k-1}P\left(A_{n}=j\right)\cdot P\left(\Pi_{i}\left(j\right)=z\right) \end{array}$$
Taking the limit from the both sides in (Equation 19) it follows
$$\begin{array}{cc} \lim_{n\to \infty } & P\left(Z_{n}=z|C_{n}=c\right)=\;\\ \; & =\lim_{n\to \infty }\sum_{j=0}^{k-1}P\left(A_{n}=j\right)\sum_{i=1}^{k!}P\left(g_{n-1}=f_{c}^{-1}\circ \Pi_{i}\right)\cdot P\left(\Pi_{i}\left(j\right)=z\right)\\ \; & =\sum_{j=0}^{k-1}P\left(A_{n}=j\right)\sum_{i=1}^{k!}P\left(\Pi_{i}\left(j\right)=z\right)\cdot \lim_{n\to \infty }P\left(g_{n-1}=f_{c}^{-1}\circ \Pi_{i}\right)=\\ \; & =\sum_{j=0}^{k-1}P\left(A_{n}=j\right)\sum_{i=1}^{k!}P\left(\Pi_{i}\left(j\right)=z\right)\cdot \frac{1}{k!}\\ \; & =\frac{1}{k!}\sum_{j=0}^{k-1}P\left(A_{n}=j\right)\sum_{i=1}^{k!}P\left(\Pi_{i}\left(j\right)=z\right)\\ \; & =\frac{1}{k!}\sum_{j=0}^{k-1}P\left(A_{n}=j\right)\cdot \left(k-1\right)!=\\ \; & =\frac{\left(k-1\right)!}{k!}\sum_{j=0}^{k-1}P\left(A_{n}=j\right)=\frac{1}{k} \end{array}$$
which proves the statement.By the definition of mutual information we have that
(20)$$I\left(Z_{n},C_{n}\right)=H\left(Z_{n}\right)-H\left(Z_{n}|C_{n}\right)$$
We will start with computing $H\left(Z_{n}\right).$
(21)$$H\left(Z_{n}\right)=\sum_{i=0}^{k-1}P\left(Z_{n}=i\right)\log_{2}\frac{1}{P\left(Z_{n}=i\right)}$$
Taking the limit from the both sides in (Equation 21) and using Theorem 1 we have
(22)$$\lim_{n\to \infty }H\left(Z_{n}\right)=\log_{2}k.$$
In the same way it follows that
(23)$$\begin{array}{cc} H\left(Z_{n}|C_{n}\right) & =\sum_{i=0}^{m-1}P\left(C_{n}=i\right)\cdot H\left(Z_{n}|C_{n}=i\right)=\\ \; & =\sum_{i=0}^{m-1}P\left(C_{n}=i\right)\cdot \sum_{j=0}^{k-1}P\left(Z_{n}=j|C_{n}=c_{i}\right)\log_{2}\frac{1}{P\left(Z_{n}=j|C_{n}=i\right)} \end{array}$$
Taking the limit from both sides in (Equation 23) and using part 1 of the Theorem 4 we obtain
(24)$$\begin{array}{cc} \; & \lim_{n\to \infty }H\left(Z_{n}|C_{n}\right)=\\ \; & =\sum_{i=0}^{m-1}P\left(C_{n}=i\right)\cdot \sum_{j=0}^{k-1}\lim_{n\to \infty }P\left(Z_{n}=j|C_{n}=i\right)\cdot \lim_{n\to \infty }\log_{2}\frac{1}{P\left(Z_{n}=j|C_{n}=i\right)}\\ \; & =\sum_{i=0}^{m-1}P\left(C_{n}=i\right)\cdot \sum_{j=0}^{k-1}\frac{1}{k}\log_{2}k=\log_{2}k\cdot \sum_{i=0}^{m-1}P\left(C_{n}=i\right)=\log_{2}k. \end{array}$$
Using (Equation 22) and( Equation 24) in ( Equation 20) it follows that
(25)$$\lim_{n\to \infty }I\left(Z_{n},C_{n}\right)=\lim_{n\to \infty }H\left(Z_{n}\right)-\lim_{n\to \infty }H\left(Z_{n}|C_{n}\right)=\log_{2}k-\log_{2}k=0$$ which proves the statement. □


**Theorem** **5.**
*Under the conditions of the Theorem 1 we have*

*$\lim_{n\to \infty }P\left(Z_{n}=z|A_{n}=a\right)=\frac{1}{k}$*

$\lim_{n\to \infty }I\left(Z_{n},A_{n}\right)=0$

*where $z,a\in I_{k}.$*


**Proof.** 
By the definition of conditional probability it follows that
(26)$$\begin{array}{cc} P & \left(Z_{n}=z|A_{n}=a\right)=\;\\ \; & =\frac{P\left(Z_{n}=z\wedge A_{n}=a\right)}{P\left(A_{n}=a\right)}=\frac{P\left(g_{n}\left(A_{n}\right)=z\wedge A_{n}=a\right)}{P\left(A_{n}=a\right)}\\ \; & =\frac{P\left(\left(\overset{k!}{\underset{i=1}{⋃}}g_{n}=\Pi_{i}\wedge \Pi_{i}\left(A_{n}\right)=z\right)\wedge A_{n}=a\right)}{P\left(A_{n}=a\right)}\\ \; & =\frac{P\left(\left(\overset{k!}{\underset{i=1}{⋃}}g_{n}=\Pi_{i}\wedge \Pi_{i}\left(A_{n}\right)=z\right)\wedge A_{n}=a\right)}{P\left(A_{n}=a\right)}\\ \; & =\frac{P\left(\overset{k!}{\underset{i=1}{⋃}}\left(\left(\left(g_{n}=\Pi_{i}\wedge \Pi_{i}\left(A_{n}\right)=z\right)\wedge A_{n}=a\right)\right)\right)}{P\left(A_{n}=a\right)}. \end{array}$$
Because $\left(\left(\left(g_{n}=\Pi_{i}\wedge \Pi_{i}\left(A_{n}\right)=z\right)\wedge A_{n}=a\right)\right)$ and $\left(\left(\left(g_{n}=\Pi_{j}\wedge \Pi_{j}\left(A_{n}\right)=z\right)\wedge A_{n}=a\right)\right)$ are disjoint events when i≠j from (Equation 26) it follows that
(27)$$\begin{array}{cc} P & \left(Z_{n}=z|A_{n}=a\right)=\;\\ \; & =\frac{P\left(\overset{k!}{\underset{i=1}{⋃}}\left(\left(\left(g_{n}=\Pi_{i}\wedge \Pi_{i}\left(A_{n}\right)=z\right)\wedge A_{n}=a\right)\right)\right)}{P\left(A_{n}=a\right)}\\ \; & =\frac{\sum_{i=1}^{k!}P\left(\left(g_{n}=\Pi_{i}\wedge \left(\Pi_{i}\left(A_{n}\right)=z\wedge A_{n}=a\right)\right)\right)}{P\left(A_{n}=a\right)}. \end{array}$$
Using that $P\left(A\wedge B\right)=P\left(A|B\right)\cdot P\left(B\right)$ from (Equation 27) it follows that
(28)$$\begin{array}{cc} P & \left(Z_{n}=z|A_{n}=a\right)=\;\\ \; & =\frac{\sum_{i=1}^{k!}P\left(\left(g_{n}=\Pi_{i}\wedge \left(\Pi_{i}\left(A_{n}\right)=z\wedge A_{n}=a\right)\right)\right)}{P\left(A_{n}=a\right)}\\ \; & =\frac{\sum_{i=1}^{k!}P\left(g_{n}=\Pi_{i}|\left(\Pi_{i}\left(A_{n}\right)=z\wedge A_{n}=a\right)\right)\cdot P\left(\Pi_{i}\left(a\right)=z\wedge A_{n}=a\right)}{P\left(A_{n}=a\right)}\\ \; & =\sum_{i=1}^{k!}P\left(g_{n}=\Pi_{i}|\left(\Pi_{i}\left(A_{n}\right)=z\wedge A_{n}=a\right)\right)\cdot \frac{P\left(\Pi_{i}\left(a\right)=z\wedge A_{n}=a\right)}{P\left(A_{n}=a\right)}\\ \; & =\sum_{i=1}^{k!}P\left(g_{n}=\Pi_{i}\;|\;\left(\Pi_{i}\left(A_{n}\right)=z\wedge A_{n}=a\right)\right)\cdot P\left(\Pi_{i}\left(A_{n}\right)=z\;|\;A_{n}=a\right). \end{array}$$
Because $\left\{g_{n}=\Pi_{i}\right\}$ and $\left\{\Pi_{i}\left(A_{n}\right)=z\wedge A_{n}=a\right\}$ are independent random variables from (Equation 28) it follows that
(29)$$\begin{array}{cc} P & \left(Z_{n}=z|A_{n}=a\right)=\;\\ \; & =\frac{\sum_{i=1}^{k!}P\left(\left(g_{n}=\Pi_{i}\wedge \left(\Pi_{i}\left(A_{n}\right)=z\wedge A_{n}=a\right)\right)\right)}{P\left(A_{n}=a\right)}\\ \; & =\frac{\sum_{i=1}^{k!}P\left(g_{n}=\Pi_{i}|\left(\Pi_{i}\left(A_{n}\right)=z\wedge A_{n}=a\right)\right)\cdot P\left(\Pi_{i}\left(a\right)=z\wedge A_{n}=a\right)}{P\left(A_{n}=a\right)}=\\ \; & =\sum_{i=1}^{k!}P\left(g_{n}=\Pi_{i}\;|\;\left(\Pi_{i}\left(A_{n}\right)=z\wedge A_{n}=a\right)\right)\cdot P\left(\Pi_{i}\left(A_{n}\right)=z\;|\;A_{n}=a\right)\\ \; & =\sum_{i=1}^{k!}P\left(g_{n}=\Pi_{i}\right)\cdot P\left(\Pi_{i}\left(a\right)=z\;\right) \end{array}$$
Taking the limits from the both sides in (Equation 29) it follows
$$\begin{array}{cc} \lim_{n\to \infty } & P\left(Z_{n}=z|A_{n}=a\right)=\\ \; & =\lim_{n\to \infty }\sum_{i=1}^{k!}P\left(g_{n}=\Pi_{i}\right)\cdot P\left(\Pi_{i}\left(a\right)=z\;\right)\\ \; & =\sum_{i=1}^{k!}\lim_{n\to \infty }P\left(g_{n}=\Pi_{i}\right)\cdot P\left(\Pi_{i}\left(a\right)=z\;\right)\\ \; & =\sum_{i=1}^{k!}\frac{1}{k!}\cdot P\left(\Pi_{i}\left(a\right)=z\;\right)\\ \; & =\frac{1}{k!}\sum_{i=1}^{k!}P\left(\Pi_{i}\left(a\right)=z\;\right)\\ \; & =\frac{1}{k!}\cdot \left(k-1\right)!=\frac{1}{k} \end{array}$$In the same way as in the Theorem 4 it follows that
$$\lim_{n\to \infty }H\left(Z_{n}\right)=\log_{2}k$$
By the definition of conditional entropy it follows that
(30)$$\begin{array}{cc} H\left(Z_{n}|A_{n}\right) & =\sum_{i=0}^{k-1}P\left(A_{n}=i\right)\cdot H\left(Z_{n}|A_{n}=i\right)\\ \; & =\sum_{i=0}^{k-1}P\left(A_{n}=i\right)\cdot \sum_{j=0}^{k-1}P\left(Z_{n}=j|A_{n}=i\right)\log_{2}\frac{1}{P\left(Z_{n}=j|A_{n}=i\right)} \end{array}$$
Taking the limit of both sides in (Equation 30) and statement of the part 1 we obtain
(31)$$\begin{array}{cc} \; & \lim_{n\to \infty }H\left(Z_{n}|A_{n}\right)=\\ \; & =\sum_{i=0}^{k-1}P\left(A_{n}=i\right)\cdot \sum_{j=0}^{k-1}\lim_{n\to \infty }P\left(Z_{n}=j|A_{n}=i\right)\cdot \lim_{n\to \infty }\log_{2}\frac{1}{P\left(Z_{n}=j|A_{n}=i\right)}\\ \; & =\sum_{i=0}^{k-1}P\left(A_{n}=i\right)\cdot \sum_{j=0}^{k-1}\frac{1}{k}\log_{2}k=\log_{2}k\cdot \sum_{i=0}^{k-1}P\left(A_{n}=i\right)=\log_{2}k \end{array}$$
And finally, using the (Equation 22) and (Equation 31) in the definition for the $I\left(Z_{n},A_{n}\right)$, it follows that
$$\lim_{n\to \infty }I\left(Z_{n},A_{n}\right)=\lim_{n\to \infty }H\left(Z_{n}\right)-\lim_{n\to \infty }H\left(Z_{n}|A_{n}\right)=\log_{2}k-\log_{2}k=0$$ which proves the statement. □


### 3.3. Periodicity

Every pseudo-random generator can be viewed as a finite automaton with output over the finite set of states and symbols. Because the automaton transition function is deterministic it follows that the output sequence must be periodic. So, ${\left\{A_{n}\right\}}_{n=1}^{\infty },$ and ${\left\{C_{n}\right\}}_{n=1}^{\infty }$ are periodic and denote their periods by *A* and *B* respectively. It is easy to verify that ${\left\{g_{n}\right\}}_{n=0}^{\infty },$ and ${\left\{Z_{n}\right\}}_{n=1}^{\infty }$ are periodic too and denote their periods G,Z respectively. In this part relations between A,C,G and *Z* are considered and some sufficient conditions under A,C,GM are defined which improves the value of *Z*. For that we need a few Lemmas.

To find out period of ${\left\{Z_{n}\right\}}_{n=1}^{\infty }$ we will first determine the period of the ${\left\{g_{n}\right\}}_{n=0}^{\infty }$.

**Lemma** **1.**
*Denote by $l\in \left\{1,2,...,k!\right\}$ the order of the permutation $\prod_{i=1}^{C}f_{C_{i}}$. Then the period of ${\left\{g_{n}\right\}}_{n=0}^{\infty }$ is lC.*


**Proof.** First we have to prove that lC is a period, but it is straightforward.
$$g_{k+\lambda lC}=f_{C_{1}}\circ f_{C_{2}}\circ \ldots \circ f_{C_{\lambda lC}}\circ f_{C_{1+\lambda lC}}\circ \ldots \circ f_{C_{k+\lambda lC}}$$
and because *C* is period of the ${\left\{C_{n}\right\}}_{n=1}^{\infty }$ we have
$$\begin{array}{cc} \; & f_{C_{1}}\circ f_{C_{2}}\circ \ldots \circ f_{C_{\lambda lC}}\circ f_{C_{1+\lambda lC}}\circ \ldots \circ f_{C_{k+\lambda lC}}=\\ \; & ={\left(f_{C_{1}}\circ f_{C_{2}}\circ \ldots \circ f_{C_{C}}\right)}^{\lambda l}\circ f_{C_{1}}\circ \ldots \circ f_{C_{k}}\\ \; & =f_{C_{1}}\circ \ldots \circ f_{C_{k}}\\ \; & =g_{k} \end{array}$$Next step is to prove that lC is the fundamental period i.e., that every other period is divisible by lC.Suppose contrary, that lC isn’t the fundamental period i.e., that the fundamental period is d,d∣lC and d<lC. From $g_{k+\lambda d}=g_{k}$ we have
(32)$$\prod_{i=k+1}^{k+\lambda d}f_{C_{i}}=I,\;k\in N,k\geq 1$$
where *I* is the identical permutation. Multiplying (Equation 32) with $f_{C_{k}}$ from the left we have
$$\prod_{i=k}^{k+\lambda d-1}f_{C_{i}}\circ f_{C_{k+\lambda d}}=f_{C_{k}}$$
and applying (Equation 32) we have
$$f_{C_{k+\lambda d}}=f_{C_{k}},\;k\geq 1.$$From this we have
$$C_{k+\lambda d}=C_{k,\;}k\geq 1$$
which means that *d* is a period of ${\left\{C_{n}\right\}}_{n=1}^{\infty }$ and that C|d i.e., d=rC,r<l. Now, look at $g_{1+\lambda d}$
$$\begin{array}{cc} g_{1+\lambda d} & =\prod_{i=1}^{1+\lambda d}f_{C_{i}}=\prod_{i=1}^{\lambda rC}f_{C_{i}}\circ f_{C_{1+\lambda rC}}=\\ \; & ={\left(\prod_{i=1}^{C}f_{C_{i}}\right)}^{\lambda r}\circ f_{C_{1}}=f_{C_{1}}=g_{1} \end{array}$$
and we conclude that
$${\left(\prod_{i=1}^{C}f_{C_{i}}\right)}^{\lambda r}=I$$
for every λ≥1. Finally we have
$${\left(\prod_{i=1}^{C}f_{C_{i}}\right)}^{r}=I$$
and conclude that l|r which is in contradiction with r<l so we proved that lC is the fundamental period of ${\left\{g_{n}\right\}}_{n=0}^{\infty }$. □

**Lemma** **2.**
*Let G be the fundamental period of ${\left\{g_{n}\right\}}_{n=0}^{\infty }$. If $\left(G,A\right)=1$ and $\left\{A_{1},A_{2},\ldots ,A_{n},\ldots \right\}=I_{k}$ then period of ${\left\{Z_{n}\right\}}_{n=1}^{\infty }$ is GA.*


**Proof.** By straightforward computation we can easily check that GA is a period of the ${\left\{Z_{n}\right\}}_{n=1}^{\infty }$. We need only to prove that GA is fundamental period of ${\left\{Z_{n}\right\}}_{n=1}^{\infty }$.Suppose that the fundamental period isn’t GA but it is *d*. Then d∣GA and because $\left(G,A\right)=1$ we have $d=d_{1}d_{2},\left(d_{1},d_{2}\right)=1$ and $d_{1}\left|G,\;d_{2}\right|A$.Further,
(33)$$g_{k+\lambda d}\left(A_{k+\lambda d}\right)=g_{k}\left(A_{k}\right)$$
for every λ≥1,λ∈N. We can set $\lambda =\lambda_{1}\cdot \frac{G}{d_{1}},\lambda_{1}\geq 1$ in the equation above which transform it to
$$g_{k+\lambda_{1}Gd_{2}}\left(A_{k+\lambda_{1}Gd_{2}}\right)=g_{k}\left(A_{k}\right)$$
and having in mind that *G* is a period of ${\left\{g_{n}\right\}}_{n=0}^{\infty }$ we obtain
$$g_{k}\left(A_{k+\lambda_{1}Gd_{2}}\right)=g_{k}\left(A_{k}\right).$$From the fact that $g_{k}$ is bijection it follows that
$$A_{k+\lambda_{1}Gd_{2}}=A_{k}$$
which means that $Gd_{2}$ is a period of ${\left\{A_{n}\right\}}_{n=1}^{\infty }$ and consequently that $A|Gd_{2}$. Using that $\left(G,A\right)=1$ we have that $A|d_{2}$ and with $d_{2}|A$ we conclude that $A=d_{2}$. According to former observations we have that *d* must be of the form $d_{1}A$ and rewriting of (Equation 33) yields
$$g_{k+\lambda d_{1}A}\left(A_{k+\lambda d_{1}A}\right)=g_{k}\left(A_{k}\right).$$This equation we can simplify to
$$g_{k+\lambda d_{1}A}\left(A_{k}\right)=g_{k}\left(A_{k}\right)$$
because *A* is a period of ${\left\{A_{n}\right\}}_{n=1}^{\infty }.$ Now we put our attention to $g_{k+nG+\lambda d_{1}A}\left(A_{k+nG+\lambda d_{1}A}\right)$.
$$\begin{array}{cc} g_{k+\lambda d_{1}A}\left(A_{k+nG}\right) & =g_{k+nG+\lambda d_{1}A}\left(A_{k+nG+\lambda d_{1}A}\right)=\\ \; & =g_{k+nG}\left(A_{k+nG}\right)=\\ \; & =g_{k}\left(A_{k+nG}\right) \end{array}$$
so we have that functions $g_{k+\lambda d_{1}A},g_{k}$ are equal on the set $\left\{A_{k+nG}\;|\;n\in N\right\}$. Set $\left\{x\;|\;k+nG\equiv_{A}x,\;n\in N\right\}$ is equal $I_{A},$ because $\left(G,A\right)=1,$ and from the periodicity of ${\left\{A_{n}\right\}}_{n=1}^{\infty }$ follows $\left\{A_{k+nG}\;|\;n\in N\right\}=I_{k}$. Functions $g_{k+\lambda d_{1}A},g_{k}$ are equal on their domain so we have that
$$g_{k+\lambda d_{1}A}=g_{k}\;,\lambda \geq 1,\lambda \in N$$
which means that $d_{1}A$ is a period of the ${\left\{g_{n}\right\}}_{n=0}^{\infty }$. In the same way as above we have that $d_{1}=G$ and we have that the fundamental period of the ${\left\{Z_{n}\right\}}_{n=1}^{\infty }$ is GA. □

Following Corollary is a trivial consequence of the Lemma 2.

**Corollary** **1.**
*If $\left(G,A\right)=1$ then the period of the ${\left\{Z_{n}\right\}}_{n=1}^{\infty }$ is greater or equal to A.*


This corollary shows that with suitably chosen pseudo-random sequence ${\left\{A_{n}\right\}}_{n=1}^{\infty }$ output sequence will have the period greater or equal to period of the ${\left\{A_{n}\right\}}_{n=1}^{\infty }.$

The next theorem, the main result of this paragraph, is a straightforward application of the former Lemmas.

**Theorem** **6.**
*Let $l\in \left\{1,2,...,k!\right\}$ denote the order of the permutation $\prod_{i=1}^{C}f_{C_{i}}$. Then if $\left(lC,A\right)=1$ and $\left\{A_{1},A_{2},\ldots ,A_{n},\ldots \right\}=I_{k},$ the period of ${\left\{Z_{n}\right\}}_{n=1}^{\infty }$ is lCA.*


A statement of this theorem is a stronger variant of the Corollary 1 and it shows that with appropriately selected pseudo-random sequences ${\left\{A_{n}\right\}}_{n=1}^{\infty }$ and ${\left\{B_{n}\right\}}_{n=1}^{\infty }$ period of the generator output sequence is significantly longer then period sequences ${\left\{A_{n}\right\}}_{n=1}^{\infty }$ and ${\left\{B_{n}\right\}}_{n=1}^{\infty }$.

## 4. Conclusions

Confidentiality of different sensor networks is a very serious requirement since such networks cannot fully achieve their purpose without having the necessary security. Namely, for various IoT applications, which typically have limited processing capabilities, restricted memory capacity and power constraints, one of the key challenges is to design an efficient and reliable cryptographic generator that meets the desired security requirements. In this paper, we defined a pseudo-random generator that can, in some way, be considered as a generalization of ideas related to RC4 because it uses time varying permutations and sequences for permutation changing and addressing output element from the current permutation are considered in a general fashion. In the paper, we analyze properties of the purposed generator. The proposed pseudo-random generator can be implemented efficiently in software and hardware, for example by using output of the multiple linear shift registers as input sequences ${\left\{A_{n}\right\}}_{n=1}^{\infty },$ and ${\left\{C_{n}\right\}}_{n=1}^{\infty }$ of the generator. The security characteristics considered in the paper potentiate application of the generator in the computational constrained environments security solutions.

In the first part of the proposed generator properties analysis, the generator output sequence probability distribution is considered. Theorem 1 establishes sufficient conditions for the generator output sequence have an asymptotically uniform probability distribution. Moreover, sufficient conditions are established for the distribution of the output sequence to have the exact uniform distribution, Remark 1 and Theorem 2. The generator output sequence uniform distribution indicates resistance of the generator to attacks based on output sequence elements prediction.

The second part of the generator analysis deala with the correlation properties of the generator output sequence in which it was shown, by Theorem 3, that the output sequence elements are asymptotically independent and accordingly no immanent remote correlations are detected, unlike to the RC4 generator (see [25]). This property indicates resistance to autocorrelation type atacks.

The third part of the analysis relates to the possibility of information leaking about the internal state of the generator, sequence ${\left\{A_{n}\right\}}_{n=1}^{\infty }$ and ${\left\{C_{n}\right\}}_{n=1}^{\infty }$, through the output sequence ${\left\{Z_{n}\right\}}_{n=1}^{\infty }$. Theorems 4 and 5 show that the amount of information that flows through an output sequence tends to zero when the length of the sequence tends to infinity. In practical terms, this means that, by rejecting the initial segment of a generated sequence of a given length, the amount of information about the state of the generator flowing into the output sequence is arbitrarily small.

In the last part of the generator analysis, the period length of the output sequence is analyzed. It has been shown that if sequences ${\left\{A_{n}\right\}}_{n=1}^{\infty }$ and ${\left\{C_{n}\right\}}_{n=1}^{\infty }$ are chosen in a suitable manner and their periods satisfy the conditions of Theorem 6, the output sequence has a significantly longer period than the sequences ${\left\{A_{n}\right\}}_{n=1}^{\infty }$ and ${\left\{C_{n}\right\}}_{n=1}^{\infty }$.

According to the performed analysis results proposed generator has provably good security characteristics.

Complexity and implementation considerations about this proposal are determined by the generation and complexity of the sequences ${\left\{C_{n}\right\}}_{n=1}^{\infty }$ and ${\left\{A_{n}\right\}}_{n=1}^{\infty }.$ Relatively weak constraints demanded for the probability distribution for the sequences ${\left\{C_{n}\right\}}_{n=1}^{\infty }$ and ${\left\{A_{n}\right\}}_{n=1}^{\infty }$ in the Theorem 1 allow implementation of the efficiently generated sequences, for example sequences generated by the multiple linear feedback shift registers.

The method described in this paper makes it possible to obtain a pseudo-random sequence with asymptotically uniform distribution and longer period using two pseudo-random sequences with irregular (non-uniform) probability distributions. Required initial conditions for two pseudo-random sequences are not serious limitations for this method because they describe natural requirements for the pseudo-random sequences, i.e., that values of their elements exhaust the set on which they are defined. An interesting question arising in this context is the speed of convergence in the Theorem 1, i.e., the number of steps after which we can use the sequence ${\left\{Z_{n}\right\}}_{n=1}^{\infty }$ as uniformly distributed. It is not possible to answer this question generally because the matrix *T* is defined by the chosen set *S* and probability distribution of the random variable *C*. Consequently, for each set *S* and random variable *C* it has to be analyzed separately. In practice, this does not make any restrictions on the application of the proposed generator because in every concrete case it is possible to compute number of transition steps to achieve representation of the limit values by the desired accuracy. 

## Figures and Tables

**Figure 1 sensors-19-05322-f001:**
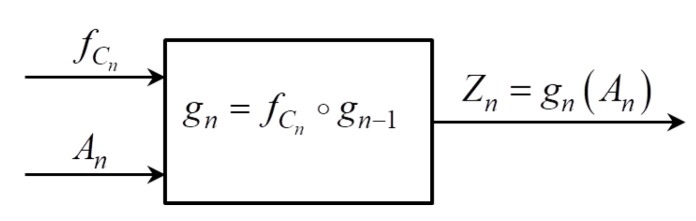
The generator graphic presentation.

**Figure 2 sensors-19-05322-f002:**
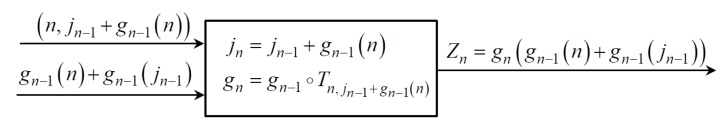
The RC4 graphic presentation, all numbers and summations assumes reduction modulo 256.

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
