# Peer review of "A Generic Model of the Pseudo-Random Generator Based on Permutations Suitable for Security Solutions in Computationally-Constrained Environments"

_sensors, 2019, doi:10.3390/s19235322_

Round 1
Reviewer 1 Report
In this paper we proposed design of the new and efficient pseudo-random generator which is parametrized by two pseudo-random sequences. The paper is well organized and the results seem to be new and the proofs are correct.
The following is a list of comments that the author should consider in revising:
The lines 79-80 on page 2, the definition of Π is not well-defined. The equation defined in eq. (1). However, it is not consist with the Figure 1. The quality of figures 1 and 2 should be improved. There are several results about the asymptotically uniform distribution. The probabilities are computed in Theorems 1-4, however, what are the constants a, b, c, and l? Please give more rigid definitions about that. Sensors (ISSN 1424-8220) provides an advanced forum for the science and technology of sensors and biosensors. This manuscript is in the scope of Cryptography. The authors should provide the relationship between the research results of this with sensors and biosensors. Can this be applied to sensors applications? Please give a case study.

Author Response
Dear Reviewer,
Thank You very much on the letter regarding the review process for the Manuscript ID: sensors-643747, Type of manuscript: Article, entitled “Generic Model of the Pseudo-random Generator Based on Permutations Suitable for Security Solutions in the Computationally Constrained Environments” by the authors: Tomislav UNKAŠEVIĆ, Zoran BANJAC, and Milan MILOSAVLJEVIĆ,.
We would like to thank you for careful and thorough reading of this manuscript and for the thoughtful comments and constructive suggestions, which help to improve the quality of this manuscript. Our response follows (blue font color).
In the lines 79-80 on page 2, the definition of П is not well-defined.The definition of П is changed by introduction of index j, j=1,2,...k! (revised manuscript page 3, line 106) The equation is defined in eq. (1). However, it is not consist with the Figure 1.
The notation of variables in Figure 1 is changed to be consistent with eq. (1) ; (revised manuscript page 3) The quality of figures 1 and 2 should be improved
The Figures 1 and 2 are reproduced with proper resolution, as is suggested. (revised manuscript, Figure 1. Page 3 and Figure 2. Page4) There are several results about the asymptotically uniform distribution. The probabilities , and are computed, however, what are the constants a, b, c, and l? Please give more rigid definitions about that.
Note of the Reviwer about constants a,b,c,l,z used in the proofs of the theorems 1-5 is correct and useful.
For Theorem 1 constant l is defined in revised manuscript line 140. page 4.
For Theorem 2 constant z is defined in the body of the Theorem 2, revised manuscript top of the page 4.
For the Theorem 3 constants a,b are defined in the body of the Theorem 3, revised manuscript line 165, page 7.
For the Theorem 4 constants z,c are defined in the body of the Theorem 4, revised manuscript line 182, page 10.
For the Theorem 5 constants z,a are defined in the body of the Theorem 5, revised manuscript line 189, page 11
For the Lemma 1 constant l is defined in the body of the Lemma 1, revised manuscript line 201, page 14.
For Theorem 6 constant l is defined in the body of the Theorem 6 revised manuscript line 213. page 16. Sensors (ISSN 1424-8220) provides an advanced forum for the science and technology of sensors and biosensors. This manuscript is in the scope of Cryptography. The authors should provide the relationship between the research results of this with sensors and biosensors. Can this be applied to sensors applications? Please give a case study.
As is suggested by the Reviewer, the motivation of the paper is more clarified and the new paragraph ( revised manuscript page 2, lines 36-62) related to the relationship between the research results with sensors in IoT applications is included. The reference list has been extended to include papers describing the need for practical implementation of efficient crypto algorithms for different sensors (references [12]- [19], [21]). which clarifies relationship between sensor networks and security.
We hope that the revised version of the paper contains the adequate answers to the valuable suggestions of the reviewer.
Thanking You in advance on Your efforts and looking forward for hearing from You soon.
Best regards,
Tomislav Unkašević

Reviewer 2 Report
This paper is interesting, well-structured and self-contained. I think that the contribution herein introduced may have several and useful practical implications. The paper appears to be sound and relies on well-known building blocks. Again, it is written by using an appropriate technical language. The mathematical notation is clearly defined and consistently used within the whole paper.
However, I think that the paper is still affected by some minor drawbacks, which need to be necessarily addressed.
- In some parts of the paper, the clarity and editorial quality of the paper weaken. As a consequence, such parts result to be quite difficult to read. Therefore, I would suggest to carefully improve the prose of writing in order to make this paper easier to read.
- The proofs of theorems are in general quite difficult to follow and understand. To overcome this limitation, I would suggest informally describing the underlying logical sketch for each proofs.
- An accurate proofreading is strongly required, since in the paper there are several typos and formatting issues, e.g., “eeq7”.
Author Response
Dear Reviewer,
Thank You very much on the letter regarding the review process for the Manuscript ID: sensors-643747, Type of manuscript: Article, entitled “Generic Model of the Pseudo-random Generator Based on Permutations Suitable for Security Solutions in the Computationally Constrained Environments” by the authors: Tomislav UNKAŠEVIĆ, Zoran BANJAC, and Milan MILOSAVLJEVIĆ,.
We would like to thank you for careful and thorough reading of this manuscript and for the thoughtful comments and constructive suggestions, which help to improve the quality of this manuscript. Our response follows (blue font color).
In some parts of the paper, the clarity and editorial quality of the paper weaken. As a consequence, such parts result to be quite difficult to read. Therefore, I would suggest to carefully improve the prose of writing in order to make this paper easier to read.
As is suggested the prose of writing is improved in order to make this paper easier to read. All equations are numbered, additional explanations are included, and connection between the proposed algorithm and practical applications is underlined. The conclusion also includes suggestions for the practical application of the proposed algorithm
The proofs of theorems are in general quite difficult to follow and understand. To overcome this limitation, I would suggest informally describing the underlying logical sketch for each proof.
For the Theorem 1 before the proof general plan of the proof is explained, revised manuscript page 4 lines 141-143. Equation (7) in first manuscript version is divided in two parts (revised manuscript equations (7) and (8)) and inference step is explained.
For the Theorem 3 proof is improved. The proof in first manuscript version is divided in four parts (revised manuscript equations (11) --(14)) with step by step explanation.
The proof of the Theorem 4 (former manuscript version, equation(11)) is divided into four parts (revised manuscript equations (16) --(19)) with corresponding guidance..
The proof of the Theorem 5 (former manuscript version, equation(18)) is divided into four parts (revised manuscript equations (26) --(29)) with appropriate explanations.
In the paper conclusion part, beside the results of the generator security analysis, practical importance of the each proved property is highlighted.
An accurate proofreading is strongly required, since in the paper there are several typos and formatting issues, e.g., “eeq7”.
The typing errors are corrected, and the denoted statements are revised, as is suggested by the Reviewer.
We hope that the revised version of the paper contains the adequate answers to the valuable suggestions of the reviewer.
Thanking You in advance on Your efforts and looking forward for hearing from You soon.
Best regards,
Tomislav Unkašević

Round 2
Reviewer 1 Report
My previuos concens have beed addressed in the new version. I have no further comments.
Reviewer 2 Report
The authors addressed all the issues I pointed out, by carefully revising the paper. The paper has been substantially improved with respect to the relative previous version. In my opinion, the paper is now ready to be accepted for publication.